# PRTGS: Precomputed Radiance Transfer of Gaussian Splats for Real-Time High-Quality Relighting

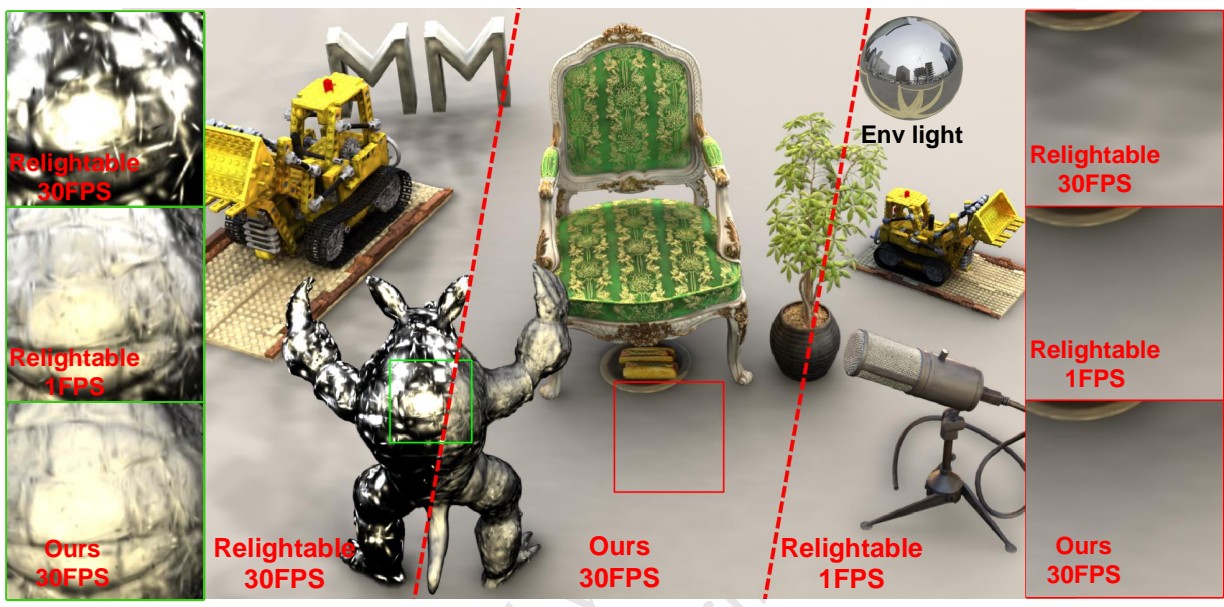

**Figure 1: An edited complex scene with over 1000000 Gaussian splats relighted by dynamic lights. (Right) Offline result (less than 1 FPS) from Relightable 3DGS [7]. (Middle) Real-time results conducted by our PRGS. (Left) Real-time results from Relightable 3DGS. All methods run equally on a Nvidia RTX 3090 GPU. Note that for comparable rendering times to the current real-time relighting method, we achieve similar quality to their offline rendering results.**

## ABSTRACT

We proposed **P**recomputed **R**adiance **T**ransfer of **G**aussian **S**plats (PRTGS), a real-time high-quality relighting method for Gaussian splats in low-frequency lighting environments that captures soft shadows and interreflections by precomputing 3D Gaussian splats' radiance transfer. Existing studies have demonstrated that 3D Gaussian splatting (3DGS) outperforms neural fields in efficiency for dynamic lighting scenarios. However, the current relighting method based on 3DGS still struggling in computing high-quality shadow and indirect illumination in real time for dynamic light, leading to unrealistic rendering results. We solve this problem by precomputing the expensive transport simulations required for complex transfer functions like shadowing, the resulting transfer functions are represented as dense sets of vectors or matrices for every Gaussian splat. We introduce distinct precomputing methods tailored for training and rendering stages, along with unique ray tracing and indirect lighting precomputation techniques for 3D Gaussian splats to accelerate training speed and compute accurate indirect lighting related to environment light. Experimental analyses demonstrate that our approach achieves state-of-the-art visual quality while maintaining competitive training times and importantly allows high-quality real-time (30+ fps) relighting for dynamic light and relatively complex scenes at 1080p resolution. **We provide a video that shows more details of our real-time rendering results under dynamic lighting conditions in supplementary materials.**

*Conference'17, July 2017, Washington, DC, USA*
© 2024 Association for Computing Machinery.
ACM ISBN 978-x-xxxx-xxxx-x/YY/MM...$15.00
https://doi.org/10.1145/nnnnnnn.nnnnnnn

## CCS CONCEPTS

• **Computing methodologies** → **Rendering**; **Ray tracing**; *Point-based models*; *Machine learning algorithms*.

## KEYWORDS

Precomputed Radiance Transfer, Radiance Field, 3D Gaussian Splatting, Relighting

# 1 INTRODUCTION

3D Gaussian Splatting [11] has garnered significant attention from the community as a promising approach for various tasks in 3D scene reconstruction. The utilization of 3DGS presents the potential for individuals to reconstruct their surrounding environment using contemporary technological devices such as smartphones and computers in minutes. Furthermore, individuals can modify their reconstructed world according to their unique preferences, which makes it particularly attractive for multimedia applications and could potentially spur innovation within the multimedia industry. The foundation for achieving this lies in a real-time and high-quality inverse rendering, relighting, and scene editing method.

The achievement of real-time inverse rendering and relighting has been a long-standing problem. Method based on Neural Radiance Fields (NeRF) [9, 20, 37] have exhibited noteworthy accomplishments in high-quality material editing, illumination computing, and shadow estimation. However, these techniques struggle with the computational overhead and cannot achieve the desired quality in dynamic environments. The integration of MLPs within these methods gives rise to inherent obstacles owing to their restricted expressive capacity and substantial computational demands. These obstacles engender a considerable curtailment of the effectiveness and efficiency of inverse rendering. Current works [7, 16, 27] introduce 3DGS to inverse rendering instead of NeRF, achieving high-performance inverse rendering and relighting by employing a set of 3D Gaussian splats to represent a 3D scene. However, they still face challenges in the real-time computation of high-quality indirect lighting and shadows in dynamic environmental lighting conditions, primarily due to the utilization of ray tracing or ambient occlusion techniques. The former struggles with balancing quality and speed because of the immense number of Gaussian splats involved [7], while the latter have difficulty in computing realistic indirect lighting [16].

In this paper, we solve the aforementioned challenge by introducing Precomputed Radiance Transfer (PRT) [28] to 3DGS. Starting with assigning each 3D Gaussian splat with normal (geometry), visibility, and BRDF attributes, we precompute the expensive transport simulation required by complex transfer functions like shadowing and interreflection for given 3d Gaussian splats. The precomputed transfer functions and incident radiance are encoded as either a dense set of vectors (diffuse cases) or matrices (glossy cases), utilizing a low-order spherical harmonic (SH) basis for each Gaussian splats distribution. This approach allows for an efficient representation of the complex transfer functions while maintaining a high level of accuracy. Leveraging the linearity inherent in light transport, we streamline the light integration process to a straightforward dot product operation between their coefficient vectors for diffuse surfaces, or a compact transfer matrix for glossy surfaces, significantly reducing computational overhead.

Our approach not only enhances real-time rendering quality in dynamic lighting but also opens up new possibilities for interactive multimedia applications. we developed unique precomputing and ray tracing methods that are specifically tailored to accommodate the unique geometry and rendering pipeline of 3DGS. This ensures that our approach is optimized to provide high-quality results while maintaining computational efficiency. Specifically,

during the training stage, we carefully designed different reflection strategies for various surfaces under different conditions, ensuring the quality of interreflection while significantly reducing training time. Our unique ray tracing method allows us to perform only one-bounce ray tracing throughout the whole training or testing stage, resulting in real-time photorealistic rendering results. Extensive experimental analyses have demonstrated that our proposed approach significantly outperforms existing methods on synthetic and real-world datasets across multiple tasks. We summarize our main contributions as follows:

- We applied PRT to 3DGS for the first time, achieving high-quality real-time relighting in complex scenes under dynamic lighting conditions, while also supporting high-quality scene editing.
- For efficiency, we designed distinct precomputing methods for both training and rendering. Additionally, we devised unique ray tracing and indirect lighting precomputation methods for 3DGS to accelerate training speed and compute accurate indirect illumination related to environmental lighting.
- Through comprehensive experimentation, we have demonstrated that our approach outperforms relevant schemes. The experimental results highlight that our method not only facilitates high-quality real-time relighting but also excels in supporting high-quality scene editing.

# 2 RELATED WORK

## 2.1 Radiance Fields

Neural Radiance Field (NeRF) [18] has arisen as a significant development in the field of Computer Vision and Computer Graphics, used for synthesizing novel views of a scene from a sparse set of images by combining machine learning with geometric reasoning. Recently, a plethora of research and methodologies built upon NeRF have been proposed. For example, [15, 22, 23, 31] extend NeRF to dynamic and non-grid scenes, [2, 3] significantly improve the rendering quality of NeRF. Recently, researchers have collectively recognized that the bottleneck in efficiency lies in querying the neural field, prompting efforts to address it. InstantNGP [19] combines a neural network with a multiresolution hash table for efficient evaluation while Plenoxels [6] replaces neural networks with a sparse voxel grid. 3D Gaussian splatting [11] further adopts a discrete 3D Gaussian representation of scenes, significantly accelerating the training and rendering of radiance fields. It has attracted considerable research interest in the field of generation [5, 34], relighting [7, 16, 27] and dynamic 3D scene reconstruction [30, 32].

## 2.2 Relighting and Inverse Rendering

Inverse rendering [25, 26] aims to decompose the image's appearance into the geometry, material properties, and lighting conditions. Most traditional methods simplified the problem by assuming controllable lighting conditions [1, 35]. Works based on Nerf explore more complex lighting models to cope with realistic scenarios and extensively utilize MLPs to encode lighting and materials properties. NeRV [29] and Invrender [38] train an additional MLP to model the light visibility. NeILF [33] expresses the incident lights as a neural incident light field. NeILF++ [36] integrates VolSDF with NeILF

and unifies incident light and outgoing radiance. TensoIR [9] introduces TensoRF representation which enables the computation of visibility and indirect lighting by raytracing. Works based on 3DGS [7, 16, 27] have significantly accelerated training and rendering, enabling real-time relighting and editing. However, these methods still face challenges in real-time high-quality indirect lighting computation, dynamic relighting, and shadow estimation.

### 2.3 Precompute Radiance Transfer

The fundamental concept of Precompute Radiance Transfer (PRT) [28] involves selecting an angular basis comprised of continuous functions, notably Spherical Harmonics (SH), and conducting all light transport operations within this domain. However, Spherical Harmonics are limited in terms of high frequencies, Sloan [17] then replace it with Haar wavelet. Kristensen [12] further extends PRT to local lighting and pan [21] extends it to dynamic scenes. Recently there has been a growing interest in using deep learning tools within traditional PRT frameworks [14, 24], and ideas from PRT have been used in the context of the radiance field [13].

## 3 PRELIMINARY

### 3.1 3D Gaussian splatting

Distinct from the widely adopted Neural Radiance Field, 3D Gaussian Splatting is an explicit 3D scene representation in the form of point clouds, where Gaussian splats are utilized to represent the structure of the scene. In this representation, every Gaussian splat $G$ is defined by a full 3D covariance matrix $\Sigma$ as well as the center (mean) $x \in R^3$.

$$G(x) = e^{-1/2(x)^T \Sigma^{-1}(x)} \tag{1}$$

The covariance matrix $\Sigma$ of a 3D Gaussian splat can be likened to characterizing the shape of an ellipsoid. Therefore, we can describe it using a rotation matrix R and a scale matrix S and independently optimize of both them.

$$\Sigma = RSS^T R^T \tag{2}$$

To project our 3D Gaussian splats to 2D for rendering, the method of splatting is utilized for positioning the Gaussian splats on the camera planes:

$$\Sigma' = JW\Sigma W^T J^T \tag{3}$$

Where J is the Jacobian of the affine approximation of the projective transformation and W is the viewing transformation. Following this, the pixel color is obtained by alpha-blending N sequentially layered 2D Gaussian splats from front to back:

$$C = \sum_{i \in N} c_i \alpha_i \prod_{j=1}^{i-1} (1 - \alpha_j) \tag{4}$$

Where $c_i$ is the color of each point and $\alpha_i$ is given by evaluating a 2D Gaussian with covariance $\Sigma$ multiplied with a learned per-point opacity.

### 3.2 Precompute Radiance Transfer

According to [10], the outgoing radiance $L_o$ at a point $x$ with normal $n$ observed by the camera in direction $\omega_o$ is given by the Rendering

Equation:

$$L_o(\omega_o, x) = L_e + \int_\Omega f(x, \omega_i, \omega_o) L_i(\omega_i, x)(\omega_i, n) V(\omega_i, x) d\omega_i \tag{5}$$

where $L_i$ corresponds to the incident light coming from direction $\omega_i$, and $f$ represents the Bidirectional Reflectance Distribution Function (BRDF) properties of the point corresponding to $\omega_i$ and outgoing direction $\omega_o$. If we assume that objects in the scene do not have self-emission. Further, we have:

$$L_o(\omega_o, x) = \int_\Omega T(x, \omega_i, \omega_o) L_i(\omega_i \cdot x) d\omega_i \tag{6}$$

where

$$T(x, \omega_i, \omega_o) = f(x, \omega_i, \omega_o)(\omega_i \cdot n) V(\omega_i, x) \tag{7}$$

corresponds to the radiance transfer. Our goal is to precompute incident light $L_i$ and radiance transfer $T$ and project them into SH domain. Any function $F(s)$ defined on the sphere $S$ can be represented as a set of SH basis functions:

$$F(s) = \sum_{l=0}^{n-1} \sum_{m=-l}^{l} f_l^m Y_l^m(s) \tag{8}$$

where n denotes the degree of SH and $Y_l^m(s)$ is a set of real basis of SH. Because the SH basis is orthonormal, the scalar function $F$ can be projected into its coefficients via the integral:

$$f_l^m = \int F(s) Y_l^m(s) ds \tag{9}$$

Then, $L_i(\omega_i)$ at $x$ defined on $\omega_i$ can be represented as:

$$L_i(\omega_i) = \sum_{l=0}^{n-1} \sum_{m=-l}^{l} l_l^m Y_l^m(\omega_i) = \sum_{j=1}^{n^2} l_j Y_j(\omega_i) \tag{10}$$

and for diffuse cases, $f(x, \omega_i, \omega_o) = \rho/\pi$ which is not related to $\omega_o$, We have:

$$T_i(\omega) = \sum_{l=0}^{n-1} \sum_{m=-l}^{l} t_l^m Y_l^m(\omega_i) = \sum_{j=1}^{n^2} t_j Y_j(\omega_i) \tag{11}$$

because $T$ is also defined only on $\omega_i$. Equation 5 can be written as:

$$L_o(x) = \sum_{p=1}^{n^2} \sum_{q=1}^{n^2} l_p t_q \int_\Omega Y_p(\omega_i) Y_q(\omega_i) d\omega_i \tag{12}$$

Considering:

$$\int_\Omega Y_p(\omega_i) Y_q(\omega_i) d\omega_i = \begin{cases} 1 & \text{if } q = p \\ 0 & \text{otherwise} \end{cases} \tag{13}$$

Then equation 12 can be written as:

$$L_o(x) = \sum_{i=0}^{n^2} l_i t_i = \vec{L} \cdot \vec{T} \tag{14}$$

where $\vec{T} = \{t_1, \ldots, t_{n^2}\}$ and $\vec{L} = \{l_1, \ldots, l_{n^2}\}$. In summary, we can project the lighting and radiance transfer to the basis to obtain $\vec{L}$ and $\vec{T}$. Rendering at each point is reduced to a dot product. For glossy cases, light transport will be obtained as a transfer matrix. Please refer to sec 4.5 and supplementary for more details.

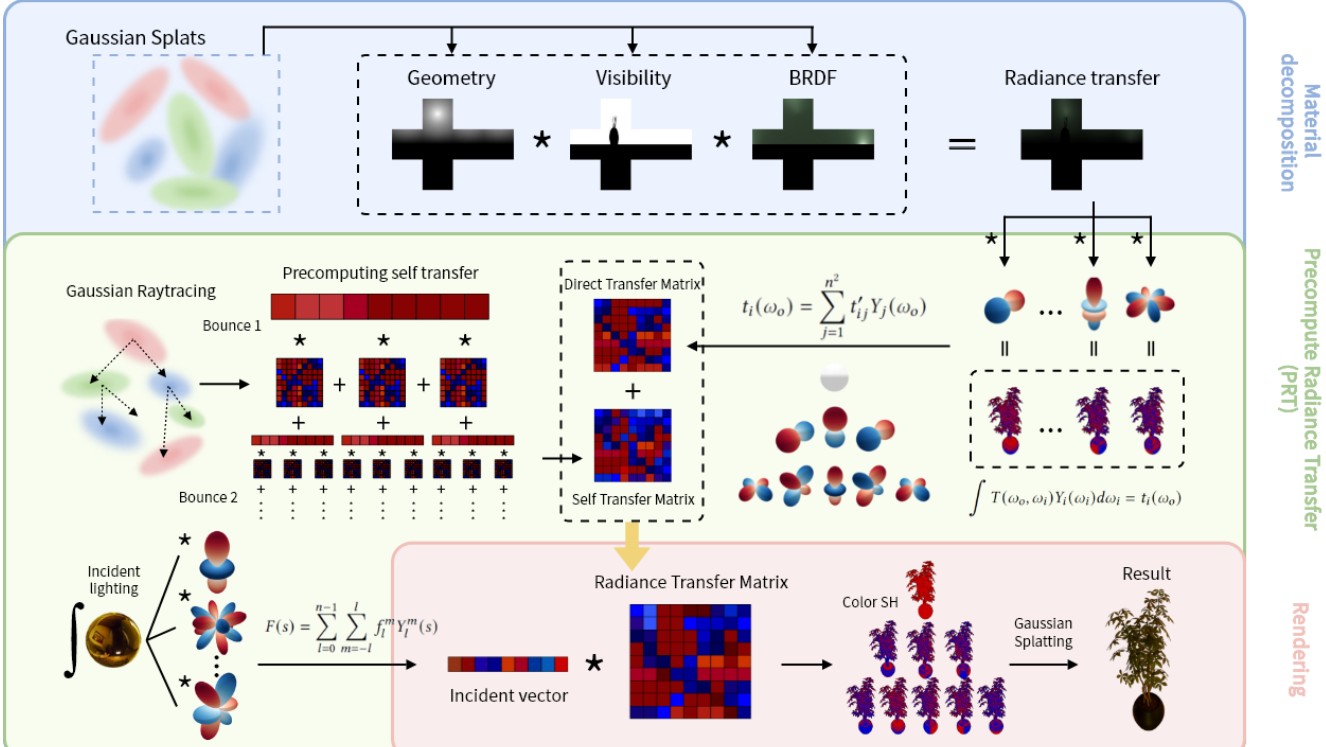

**Figure 2: The proposed rendering pipeline. Starting with a collection of 3D Gaussian splats that embody geometry, visibility, and BRDF attributes along with incident lighting, We first compute the radiance transfer for every Gaussian splat by executing equation 28, the direct transfer matrix by equation 40 and the incident vector by equation 10. Following this, we conduct one-bounce 3D Gaussian ray tracing (Sec 4.4) to get the index matrix and precompute self-transfer for every Gaussian splat recursively to estimate indirect illumination based on the index matrix. Finally, we conduct a straightforward dot product between the radiance transfer matrix and incident vector in the spherical harmonics domain and compute the final rendering result with Gaussian splatting. Note that during the testing stage, only dot product and Gaussian splatting (red background) are executed iteratively while the rest are precomputed once. All stages except ray tracing are differentiable for training.**

## 4 METHOD

### 4.1 Overview

In this section, we present our precomputed Gaussian splats radiance transfer framework, shown in Fig 2, which decomposes geometry, materials, and illumination for Gaussian splats (sec 4.2 and 4.3). In addition, we developed unique precomputing and ray tracing methods that are specifically tailored to accommodate the unique geometry and rendering pipeline of 3DGS (sec 4.4). Finally, we recursively compute self-transfer and project radiance transfer into SH coefficients for fast rendering (sec 4.5).

### 4.2 BRDF Rendering

To facilitate the physically based rendering of 3D Gaussian splats, we introduce a parametrization scheme that includes an additional set of terms for optimization purposes, *i.e.* albedo $\rho \in [0, 1]$, roughness $r \in [0, 1]$ and metallic $m \in [0, 1]$. According to Disney BRDF model [4], the BRDF property $f(\omega_i, \omega_o)$ of a material can be decomposed into two components: roughness BRDF $f_d$ and specular

BRDF $f_s$.

$$f(\omega_i, \omega_o) = (1 - m)\frac{\rho}{\pi} + \frac{DFG}{4(\omega_i \cdot n)(\omega_o \cdot n)} = f_d + f_s \quad (15)$$

where D is the microfacet distribution function, F is the Fresnel reflection and G is the geometric shadowing factor all of which are related to the roughness $r$. Then equation 5 can be written as

$$L_o(\omega_o) = L_o^s(\omega_o) + L_o^d \quad (16)$$

where:

$$L_o^d = f_d \int_\Omega L_i(\omega_i, x)(\omega_i, n)d\omega_i \quad (17)$$

$$L_o^s(\omega_o) = \int_\Omega f_s(x, \omega_i, \omega_o)L_i(\omega_i, x)(\omega_i, n)d\omega_i \quad (18)$$

### 4.3 Lighting and Geometry Modeling

**Lighting Modeling** The majority of existing methods decompose the incident light $L_i$ at point x into two components: direct illumination $L_i^{dir}$ and indirect illumination $L_i^{in}$. Equation 6 can be further

written as:

$$L_o(\omega_o) = \int_\Omega T(\omega_i, \omega_o)(L_i^{dir}(\omega_i) + (L_i^{in}(\omega_i)))d\omega_i \quad (19)$$

However, computing accurate indirect illumination in real-time for 3D Gaussian splats is a challenging task. We precompute self-transfer instead of directly computing indirect illumination. Considering:

$$L_i^{in}(-\omega_i) = \int_\Omega T^1(\omega_i^1, -\omega_i)(L_i^{dir}(\omega_i^1) + (L_i^{in}(\omega_i^1)))d\omega_i^1 \quad (20)$$

where $T^1(\omega_i^1, -\omega_i)$ is the light transfer at point (splat) $x^1$ which is hit by one-bounce ray tracing. Recursively, we have:

$$L_o(\omega_o) = \int_\Omega T'(\omega_i, \omega_o)L_i^{dir}(\omega_i)d\omega_i \quad (21)$$

where

$$T'(\omega_i, \omega_o) = T(\omega_i, \omega_o) + T(\omega_i, \omega_o)\int_\Omega T^1(\omega_i^1, -\omega_i)d\omega_i^1 \quad (22)$$

$$+ \cdots + T(\omega_i, \omega_o)\int_\Omega \cdots \int_\Omega T^n(\omega_i^n, -\omega_i^{n-1})d\omega_i^n$$

and the global direct light term $L_{dir}$ is parameterized as a globally shared SH, denoted as $\vec{l}_{dir}$, and indirect light is represented as the direct light multiplied by self-transfer.

**Geometry Modeling**   Same as [7, 9, 16, 37], we utilize the depth gradient to derive pseudo-normals, which in turn serve as guidance for optimizing normals within the 3D Gaussian splats. Given the distance from the corresponding 3D Gaussian splat to the image plane $d_i$ and the $\alpha_i$ by evaluating a 2D Gaussian with covariance $\Sigma$ multiplied with a learned per-point opacity, we obtain pseudo depth D derived from equation 4:

$$D = \sum_{i \in N} d_i \alpha_i \prod_{j=1}^{i-1}(1-\alpha_j) \quad (23)$$

The pseudo-normal $N' \in HxW$ can be computed from D. However, calculating the normal $n_i$ for each Gaussian from $N$ through interpolation is highly inaccurate. Therefore, like [7, 9], we initialize a random normal ni for each Gaussian splat and constrain it using normal loss $L_{normal}$. Here:

$$L_{normal} = \|N - N'\| \quad (24)$$

and

$$N = \sum_{i \in N} n_i \alpha_i \prod_{j=1}^{i-1}(1-\alpha_j) \quad (25)$$

## 4.4  3D Gaussian Raytracing

In the context of 3D Gaussian splats, the computational overhead of ray tracing is notably increased. Because ray tracing cannot be initiated directly from screen space, multiple sampling is required for each Gaussian splat to achieve noise-free results. Moreover, rays need to bounce recursively between Gaussian splats, as the number of bounces increases, the computation for tracing and rendering grows exponentially with the sample amount. In this paper, we present a novel ray tracing technique integrated with self-transfer precomputation. Our approach streamlines the process by doing one-bounce ray tracing during the whole training or

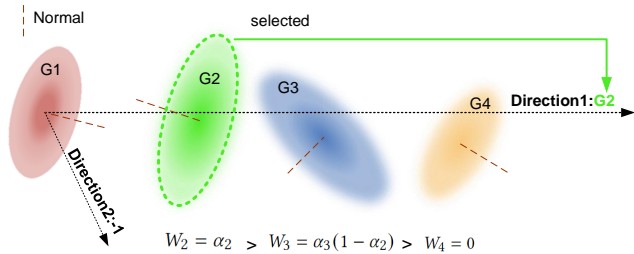

**Figure 3: 3D Gaussian raytracing. Ray from $G_1$ at direction 1 hits 3 Gaussian splats $G_2$, $G_3$ and $G4$, we compute the weight for each Gaussian splat by equation 26 and select the Gaussian splat with the biggest weight ($G_2$). Note that $W_4 = 0$ because $n_4 \cdot d_1 < 0$ and direction 2 are labeled as -1 because the ray at this direction doesn't hit any Gaussian splat.**

rendering stage and updating self-radiance transfer iteratively and recursively. As a result, we achieve precise computation of indirect illumination for multiple bounces under diverse dynamic environment lighting. Like [5, 7, 16], our proposed ray tracing technique on 3D Gaussian splats is constructed upon the Bounding Volume Hierarchy (BVH), facilitating efficient querying of visibility along a ray. Unlike conventional ray tracing methods, our approach does not involve computing irradiance during the ray tracing process. Instead, we solely compute the weights $W_i$ of the Gaussian splat intersected along each direction and record the index of the Gaussian splat with the maximum weight for each direction. Same as 4, our weight can be written as:

$$W_i = \alpha_i \prod_{j=1}^{i-1}(1-\alpha_j)max(-r_d \cdot n_i, 0) \quad (26)$$

where $r_d$ is the normalized ray direction and $n_i$ is the normal of hit Gaussian splat and $max(-r_d \cdot n_i, 0)$ prevents one Gaussian splat from being illuminated by the backside of another Gaussian splat. $\alpha_i$ is given by evaluating a 3D Gaussian splat with covariance $\Sigma$ multiplied with a learned per-splat opacity.

$$\alpha_i = \frac{r^T \Sigma r_d}{r_d^T \Sigma r_d}\sigma_i \quad (27)$$

where $\sigma_i$ is the opacity of hit Gaussian. To avoid illumination from other Gaussian splats on the same surface, we filter out all points when $\prod_{j=1}^{i-1}(1-\alpha_j) > t$ and $t$ is a hyperparameter that can be adjusted. Although we only performed one-bounce ray tracing, we can recursively compute n-bounce indirect illumination using equation 22, with minimal additional computational cost in terms of time. please refer to section 4.5 for a detailed explanation of this. In summary, we performed one-bounce ray tracing in n directions for each Gaussian splat and generated an index matrix. This matrix records the index of other Gaussian splats that exert the greatest contribution on the current Gaussian splat along n directions (if no splat is hit along a direction, it is recorded as -1). This index matrix is utilized for computing self-transfer. Note that we no longer update the positions and other geometric properties of Gaussian splats after ray tracing. Our training stage is performed on a set of

stable Gaussian splats trained by using [11] but without additional properties such as albedo.

## 4.5 Precomputing Radiance Transfer for 3D Gaussian Splats

**Transfer Vector for Training** During training, it is necessary to iteratively update properties such as roughness and metallic for Gaussian splats. Consequently, radiance transfer for Gaussian splats needs to be computed in each iteration with a known view direction. However, iteratively computing the transfer matrix is costly. Therefore, we make a slight compromise on the accuracy of indirect illumination to substantially reduce training time. Given a view direction $v$, radiance transfer at Gaussian splat x can be rewritten as:

$$T(x, v, \omega_i) = (f_s(\omega_i, v) + f_d)(v \cdot n)V(v, x) \qquad (28)$$

which is defined on sphere $\omega_i$ since $v$ is given. According to equation 11, we can simply precompute $T(x, \omega_i)$ into a transfer vector $\vec{T}$. However, the previously mentioned approach is not applicable for computing self-transfer. This is due to the inconsistency in the $\omega_o$ for interreflection, leading to exponentially increasing computational complexity. By assuming that all indirect illumination originates from diffuse surfaces, we can streamline complicated matrix calculations into vector calculations. Additionally, we precompute the diffuse radiance transfer:

$$T_{diffuse}(x, \omega_i) = f_d(\omega_i \cdot n)V(\omega_i, x) \qquad (29)$$

into $\vec{T}_{diffuse}$. In sec 4.4, we precompute the index of other Gaussian splats $x'$ that exert the greatest contribution on the current Gaussian splat along direction $d_i$, therefore, we can quickly query its corresponding diffuse radiance transfer $\vec{T}_{diffuse}^{d_i}$. According to equation 22, one-bounce self-transfer can be written as:

$$T^1 = f(d_i, v)(d_i \cdot n) \int_\Omega T_{diffuse}^{d_i}(\omega_i^1, -d_i)d\omega_i^1 \qquad (30)$$

and

$$T_{diffuse}^{d_i}(\omega_i^1) = \sum_{j=1}^{n^2} t_j Y_j(\omega_i^1) = \vec{T}_{diffuse}^{d_i} \cdot \vec{Y}(\omega_i^1) \qquad (31)$$

then:

$$T^1 = \sum_{d_i} f(d_i, v)(d_i \cdot n)\vec{T}_{diffuse}^{d_i} \cdot \vec{Y}(\omega_i^1) \qquad (32)$$

and one-bounce diffuse self-transfer can be written as

$$T_{diffuse}^1 = \sum_{d_i} \frac{\rho}{\pi}(d_i \cdot n)\vec{T}_{diffuse}^{d_i} \cdot \vec{Y}(\omega_i^1) \qquad (33)$$

Recursively, we can compute n-bounce self-transfer.

$$T^n = \sum_{d_i} f(d_i, v)(d_i \cdot n)\vec{T}_{diffuse}^{d_i, n-1} \cdot \vec{Y}(\omega_i^n) \qquad (34)$$

Finally, the total self-transfer vector is:

$$\vec{T} = \sum_{j=1}^n \vec{T}^j \qquad (35)$$

Although our method may not address specialized light paths (such as SDS path), it can efficiently compute simple specular interreflections (such as SD path) that [7, 16, 27] can't since current Gaussian splat is not assumed to be diffuse.

**Transfer Matrix for unknown direction** During relighting and other test tasks, we can precompute the transfer matrix for glossy cases since BRDF and other properties are certain. Given:

$$T(x, \omega_i, \omega_o) = (f_d + f_s(\omega_i, \omega_o))(\omega_i \cdot n)V(\omega_i, x) \qquad (36)$$

According to equation 9, we have:

$$t_i(\omega_o) = \int T(\omega_o, \omega_i)Y_i(\omega_i)d\omega_i \qquad (37)$$

$t_i(\omega_o)$ is defined on the sphere $\omega_o$, so it can be represented as another set of basis functions:

$$t_i(\omega_o) = \sum_{j=1}^{n^2} t'_{ij}Y_j(\omega_o) \qquad (38)$$

and $T(\omega_o, \omega_i)$ can be represented as:

$$T(\omega_o, \omega_i) = \sum_{i=1}^{m^2}\sum_{j=1}^{n^2} t'_{ij}Y_j(\omega_o)Y_i(\omega_i) \qquad (39)$$

where:

$$t'_{ij} = \sum_{l=1}^q\sum_{k=1}^p Y_i(\omega_i^l)T(\omega_i^l, \omega_o^l)Y_j(\omega_o^k) \qquad (40)$$

Here we sample $\omega_i$ m times and $\omega_o$ p times. Then equation 12 can be written as:

$$L_o(x) = \sum_{i=0}^{m^2}\sum_{j=0}^{n^2} l_i t'_{ij} = \vec{L} \cdot T_{m \times n} \qquad (41)$$

Like 34, we can precompute self-transfer by:

$$t'_{ij}{}^n = \sum_k \alpha(f(d_i, v)(d_i \cdot n))(t'_{kj})_{d_i}^{n-1} \cdot Y_k(-d_i)Y_j(d_i) \qquad (42)$$

However, we still opt for diffuse self-transfer in most cases due to its faster computation. Please refer to supplementary materials for more details.

## 5 EXPERIMENT

### 5.1 Implementation Details

**Dataset and Metric** We conduct experiments using benchmark datasets of TensorIR Synthetic [9], DTU [8] and Mipnerf-360 [3] to evaluate our method's performance on both synthetic and real-world scene. We evaluate the synthesized novel view and relighting results in terms of Peak Singal-to-Noise Ratio (PSNR), Structural Similarity Index Measure (SSIM), and Learned Perceptual Image Patch Similarity (LPIPS).

**Baselines** Considering the popularity and performance, we selected NeRFactor [37], NeRFdiffRec [20], Invrender [38], TesnorIR [9], Relightable 3DGS [7] and GSIR [16] as our main competitor. We conducted a comprehensive comparison between the aforementioned methods and our approach in terms of efficiency and quality. Our method are conducted on a single NVIDIA GeForce RTX 3090 GPU.

**Table 1: Quantatitive Comparison on TensoIR Synthetic dataset. Our method outperforms both previous offline and real-time methods on Novel view synthesis. Our relighting results rank first in all real-time methods and second in all methods, only behind TensoIR. Importantly, the average training time of our PRTGS is accelerated by a factor of 25x, and the average training time is accelerated by a factor of 10000x compared to TensoIR, making its performance acceptable and further demonstrating the effectiveness of our approach. The best results are marked in red, the second best are marked in blue.**

| Category | Method | Novel View Synthesis | | | Relighting | | | Render time | Train time |
|---|---|---|---|---|---|---|---|---|---|
| | | PSNR↑ | SSIM↑ | LPIPS↓ | PSNR↑ | SSIM↑ | LPIPS↓ | | |
| Offline | NeRFactor | 24.740 | 0.916 | 0.114 | 23.606 | 0.902 | 0.122 | >100s | days |
| | InvRender | 25.879 | 0.928 | 0.088 | 22.754 | 0.892 | 0.104 | 63.49s | hours |
| | TensoIR | 34.540 | 0.976 | 0.039 | **29.127** | **0.955** | **0.065** | >100s | hours |
| RealTime | NVDiffrec | 28.617 | 0.958 | 0.051 | 20.149 | 0.877 | 0.083 | **0.005s** | hours |
| | Gsir | 35.739 | 0.975 | **0.035** | 25.308 | 0.884 | 0.096 | - | minutes |
| | Relightable | **39.204** | **0.984** | 0.059 | 27.017 | 0.893 | 0.083 | 0.022s | minutes |
| | Ours | **41.985** | **0.988** | **0.022** | **27.76** | **0.903** | **0.074** | **0.013s** | minutes |

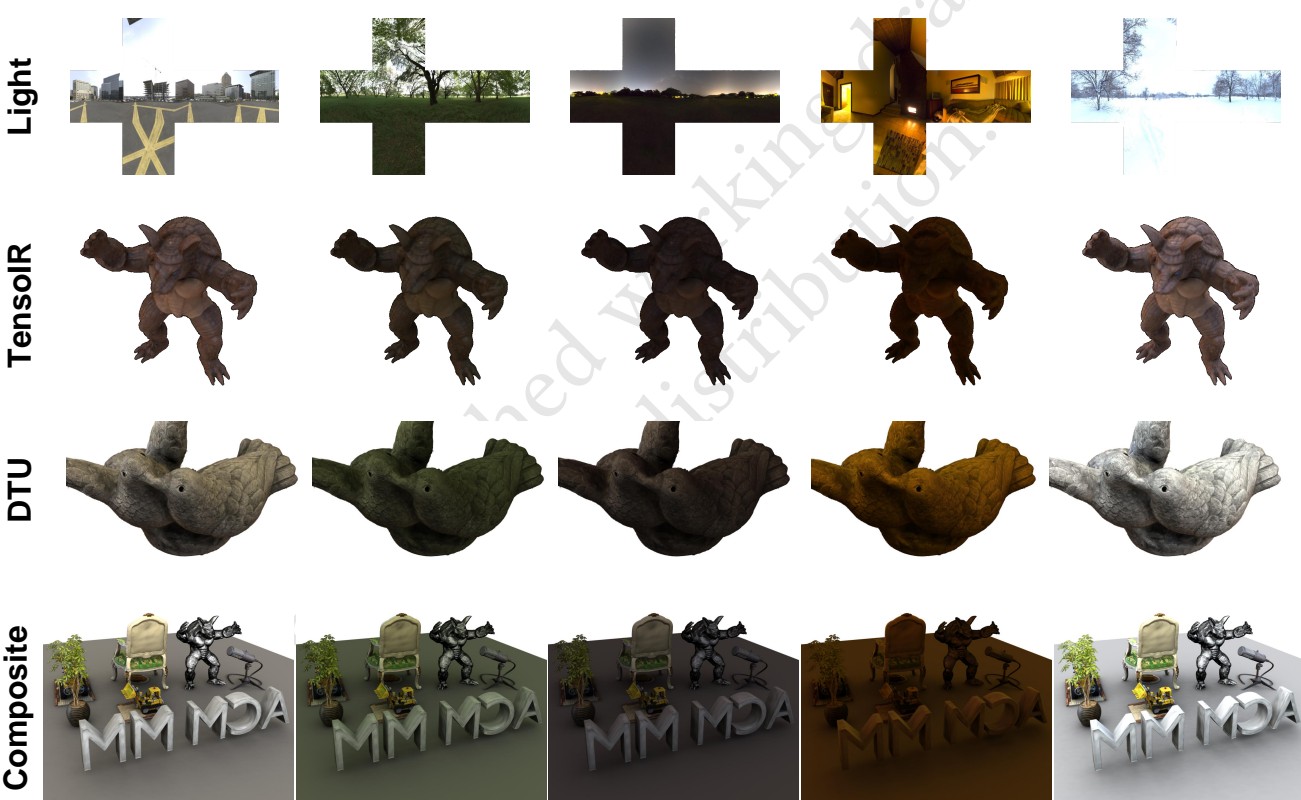

**Figure 4: High-quality relighting results achieved by our proposed method on TensoIR dataset [9], DTU dataset [8] and a composite scene created by us.**

## 5.2 Results on Synthetic Datasets

We compare our method with previous state-of-the-art offline and real-time relighting methods. Table 1 shows that our method outperforms both previous offline and real-time methods on Novel view synthesis. Our relighting results rank first in all real-time methods and second in all methods, only behind TensoIR. Taking both efficiency and quality into account, our approach achieves optimality. Figure 4 shows visual results on Relighting tasks. We test our method under different lighting conditions and our method performs well on all lighting conditions with visually appearing results. Furthermore, we also test our method on composite synthetic scenes like [7] and the result shows that our method can not only facilitate material editing like [9, 16, 27] but also support scene editing and generate photo-realistic results and soft shadow that [5] can't.

**Table 2: Quantatitive Comparison on Mip-NeRF 360. Our method surpasses most NeRF variants and Gaussian inverse rendering method dedicated to novel view synthesis**

| Method | PSNR↑ | SSIM↑ | LPIPS↓ |
|---|---|---|---|
| NeRF++ | 26.214 | 0.659 | 0.348 |
| Plenoxels | 23.625 | 0.670 | 0.443 |
| INGP-Base | 26.430 | 0.725 | 0.339 |
| INGP-Big | 26.750 | 0.752 | 0.299 |
| Gsir | 26.659 | 0.815 | 0.229 |
| Ours | **28.116** | **0.865** | **0.182** |

## 5.3 Results on real-world Datasets

We extend our evaluation to real-world dataset [3, 8] with geometric intricacies inherent and complex lighting conditions. Due to the lack of data under varying lighting conditions, Tab 2 only presents the quantitative comparisons on real-world datasets on Novel view synthesis tasks. From Tab 2 we can conclude that our real-time relighting approach even surpasses most NeRF variants and the advanced inverse rendering method dedicated to novel view synthesis. Figure 4 demonstrates reconstructed scene details including high-frequency appearance, rendering high-fidelity appearance, and recovering fine geometric details.

## 5.4 Ablation

**Indirect illumination** To demonstrate the effectiveness of our indirect illumination model, we conducted comparisons with two alternative variants: a model without indirect illumination (w/o indirect) and a model with ambient occlusion indirect illumination (AO indirect). We report the average scores on the TensoIR dataset, as outlined in Tab 3. Our analysis reveals that accurate indirect illumination plays a pivotal role in estimating accurate material decomposition and producing photorealistic rendering results. To prove that our raytracing and indirect illumination estimation method can compute accurate indirect illumination related to environment light, we compare our method with Relightable 3DGS [7] in different lighting conditions. Figure 5 shows that the indirect illumination generated by our method not only adapts to lighting conditions but also adjusts with changes in the scene.

**Table 3: Analyses on the impact of indirect illumination. Our indirect illumination method improves the rendering quality.**

| Method | TensoIR Synthetic | | | DTU | | |
|---|---|---|---|---|---|---|
| | PSNR↑ | SSIM↑ | LPIPS↓ | PSNR↑ | SSIM↑ | LPIPS↓ |
| AO indirect | 38.681 | 0.981 | 0.024 | 29.192 | 0.933 | 0.108 |
| w/o indirect | 39.122 | 0.983 | **0.022** | 29.301 | 0.934 | 0.108 |
| Ours | **41.985** | **0.988** | **0.022** | **29.458** | **0.935** | **0.107** |

**Spherical Harmonics Order** We investigate the impact of Spherical Harmonics Order on the relighting quality. We select orders 2, 3, and 6 and report the quantitative results on Tap 4. Our analysis indicates that under relatively smooth environmental lighting conditions, the order of spherical harmonic functions has

**Table 4: Analyses on the impact of SH order. SH order has little influence on relighting quality.**

| Method | PSNR↑ | SSIM↑ | LPIPS↓ | Render time |
|---|---|---|---|---|
| SH Order 2 | 27.730 | 0.904 | 0.073 | 0.012 |
| SH Order 3 | 27.722 | 0.904 | 0.073 | 0.013 |
| SH Order 6 | 27.720 | 0.904 | 0.073 | 0.050 |

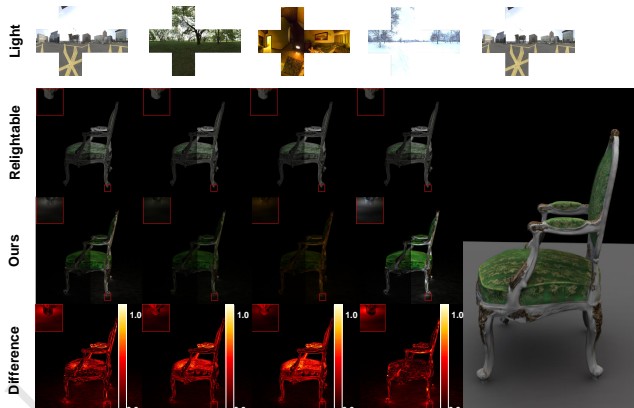

**Figure 5: Qualitative comparison on TensoIR Synthetic dataset. We visualize the indirect illumination in different lighting conditions. To showcase more details, we have merged the doubled-scaled indirect illumination results (on the right half) with the original brightness indirect illumination results (on the left half). The top-left corner exhibits a magnified view of the local area. We utilized an edited scene (Right). In contrast to other methods, our indirect illumination not only aligns well with ambient light but also accommodates the edited scene effectively (noticeable indirect illumination around the chair legs).**

a minor impact on the quality of results but a significant effect on rendering speed. Hence, to ensure efficiency, we adopt lower orders of spherical harmonic functions (2 or 3).

## 6 CONCLUSION

We proposed PRTGS, a real-time high-quality relighting method in low-frequency lighting environments for 3D Gaussian splatting. In terms of implementation, we precompute the expensive transport simulations required for complex transfer functions into sets of vectors or matrices for every Gaussian splat. We introduce distinct precomputing methods tailored for training and rendering stages, along with unique ray tracing and indirect lighting precomputation techniques for 3D Gaussian splats to accelerate training speed and compute accurate indirect lighting related to environmental light. Extensive experiments demonstrate the superior performance of our proposed method across multiple tasks, highlighting its efficacy and broad applicability in relighting, scene reconstruction and editing.

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
