# OpenReview forum: "PRTGS: Precomputed Radiance Transfer of Gaussian Splats for Real-Time High-Quality Relighting"
_acmmm.org/ACMMM/2024/Conference — MM2024 Poster_

### Official Review · Reviewer_ZMwu · 2024-05-22

**Rating:** 3
**Confidence:** 3

**Summary:**

In this paper, the authors proposed PRTGS for high-quality real-time relighting in 3DGS scenes. Experimental results show that the proposed PRTGS outperformed relevant schemes.

**Strengths:**

In this paper, the authors introduced the classical precomputed radiance transfer technique to 3DGS relighting, and designed distinct precomputing methods for efficient training and rendering.

**Limitations:**

However, some limitations remain in the submitted paper. The organization of the paper is confusing. Sec. 4 fails to adequately break down and discuss the rendering pipeline shown in Fig. 2. The derivation steps of the formulas are redundant. They should be placed in supplementary materials to make it easier for readers without relevant backgrounds to understand. Moreover, in the proposed method, how to ensure the accuracy of the pseudo-normal $N'$. This is important but omitted in the submitted paper.

According to the reported qualitative comparison (Fig. 5), the improvement of the proposed method seems trivial. In Tab. 3, without "indirect illumination", the performance decrease of "w/o indirect" also seems trivial, compared with the proposed method. In addition, the authors proposed different precomputing methods for training and testing, which should be considered to construct the ablation study.

**Suitability:**

2

---

### Official Review · Reviewer_s3oq · 2024-05-24

**Rating:** 4
**Confidence:** 2

**Summary:**

This paper proposes a novel real-time relighting method for Gaussian splatting. Specifically, it is the first to apply Precomputed Radiance Transfer (PRT) to 3D Gaussian Splatting and introduces precomputation techniques of ray tracing and indirect lighting tailored for 3DGS. Furthermore, it proposes different modeling methods of radiance transfer for training and rendering. Experimental results demonstrate that the proposed method achieves good quality performance with reduced rendering time under dynamic lighting scenarios.

**Strengths:**

1.	This paper demonstrates its novelty by introducing PRT in dynamic relighting of 3DGS and designing unique precomputing techniques for 3DGS.
2.	The theory is well-founded, and the argumentation is detailed.
3.	This paper achieves high-quality rendering results under dynamic lighting compared with the baseline Relightable.

**Limitations:**

1.	It would be better to provide the relighting ground truth in the visualization. It is hard to compare the visualized relighting results of Relightable and the proposed method without ground truth.
2.	It would be better to compare with Relightable on real-world Datasets in Table 2.
3.	The presentation of this paper can be improved. It is not so easy to follow without explanations or motivations for some equations.
4.	Please check typos in: page 2 line 168, page 3 line 283, page 5 line 502. There are reused symbol f and symbols l,t,T  without denotations in section 3.2.

**Suitability:**

3

---

### Official Review · Reviewer_E9SV · 2024-05-25

**Rating:** 5
**Confidence:** 3

**Summary:**

The paper introduces Precomputed Radiance Transfer of Gaussian Splats (PRTGS), a method for real-time, high-quality relighting of 3D Gaussian splats in low-frequency lighting environments. It addresses the shortcomings of existing 3D Gaussian Splatting methods by precomputing radiance transfer to enhance shadow quality and indirect illumination. The approach leverages spherical harmonics and unique ray tracing techniques, demonstrating SOTA visual quality and improved computational efficiency on both synthetic and real-world datasets.

**Strengths:**

1. The integration of Precomputed Radiance Transfer with 3D Gaussian splatting is a novel approach that significantly enhances real-time relighting capabilities by precomputing complex transfer functions like shadowing and interreflection.
2. The paper includes a comprehensive evaluation against SOTA methods and demonstrates the superiority of the proposed method.

**Limitations:**

1. The paper would benefit from a clearer illustration of the additional computational overhead introduced by the precomputation process. Detailed explanations of the time and resource demands specific to the pre-computation phase would aid in understanding the trade-offs involved in adopting this method for real-time applications

2. The results presented in Table 1 would be more informative if exact training times were provided, rather than broad categories such as minutes, hours, or days. This is particularly important when comparing with real-time methods, as the additional computational overhead introduced by the precomputation operation is not clearly illustrated with the current vague time frames.

3. It is unclear why the order of Spherical Harmonics has minimal impact on the relighting quality. Could this observation be attributed to the limited variety of datasets evaluated? Additionally, the paper lacks specific details regarding which datasets were used in the experiments that led to the results presented in Table 4.
·
4. The authors are encouraged to enhance their discussion on:
   a. Claims of novelty: It is essential to articulate clearly what distinguishes this work from other relighting methods in terms of visual quality. While the precomputation significantly reduces computational overhead during runtime, there appears to be no specific optimization technique employed to improve visual quality directly. The innovation seems to derive from applications of existing concepts rather than new methodologies.
   b. The additional storage burden incurred by storing the precomputed Radiance Transfer data.

Minor:
There is a typo "Tap" in the sentence "We select orders 2, 3, and 6 and report the quantitative results on Tap 4.".

**Suitability:**

3

---

### Meta-Review · Area_Chair_vbNV · 2024-07-02

**Recommendation:** Accept (Poster)
**Confidence:** 4

**Metareview:**

This paper was reviewed by three experts in the field. The recommendations are mixed, including Weak Accept, Borderline Accept, and Borderline Reject. The authors have addressed most of the concerns from the reviewers. The main remaining concern is the marginal improvement from the proposed method. After carefully examining the authors’ feedback and experimental results, ACs feel that the proposed methods do bring non-trivial improvement in illumination rendering, compared to baselines. Still, we do think the authors should show a more comprehensive ablation study to illustrate the effectiveness of the proposed solution compared to w/o indirect or AO indirect.

Based on this, and also the fact that the proposed system significantly accelerates the rendering process, we believe this work contains enough contribution and brings many useful insights to the community. Based on this, the decision is to recommend the paper for acceptance to ACM Multimedia 2024. We congratulate the authors on the acceptance of their paper!